# Modifying Anti-Compression Property and Water-Soluble Ability of Polyglycolic Acid via Melt Blending with Polyvinyl Alcohol

**DOI:** 10.3390/polym14163375

**Published:** 2022-08-18

**Authors:** Liao Wei, Shuyue Ma, Mengyuan Hao, Lanrong Ma, Xiang Lin

**Affiliations:** 1State Key Laboratory of Shale Oil and Gas Enrichment Mechanisms and Effective Development, SINOPEC Research Institute of Petroleum Engineering Co., Ltd., Beijing 102206, China; 2College of Mechanical and Electrical Engineering, Beijing University of Chemical Technology, Beijing 100029, China; 3School of Chemistry and Biological Engineering, University of Science and Technology Beijing, Beijing 100083, China

**Keywords:** polyglycolic acid, thermal stability, water-soluble ability, mechanical property

## Abstract

Biodegradable polymeric materials have become the most attractive research interest in recent years and are gradually widely used in various fields in the case of environmental pollution. In this paper, binary blends, mainly including varying contents of polyglycolic acid (PGA) and poly(vinyl alcohol) (PVA), were prepared via a melt compounding strategy. The ethylene-methyl acrylate-glycidyl methacrylate (EMA-GMA) was employed as the compatibilizer to improve the compatibility between the PGA and PVA and the polyolefin elastomer (POE) was used as toughening agent. The anti-compression property and water-soluble ability of the blends were particularly studied to explore their potential application in an oil/gas exploitation field. Special attentions were paid to the evolution of the water-soluble ability of PGAX with the PVA concentration. Furthermore, isothermal shear measurement and thermogravimetric analysis were performed to evaluate the thermal stability of PGA and PGA blends (PGAX) during melt processing. The results showed that the incompatibility between PGA and PVA largely deteriorated the mechanical property, i.e., anti-compression strength, leading to fragile characteristics under a lower compressive load for the PGAX samples with varied contents of PVA. The presence of PVA and EMA-GMA greatly enhanced the viscoelasticity of the PGA melt, showing an increased storage modulus and viscosity at a low shear frequency; however, the thermal instability of PGAX was intensified owing to the greater ease of thermal degradation of PVA than that of PGA. Meanwhile, the water-soluble ability of PGAX was improved due to the high water dissolution of PVA, which played the role as a sacrificial material. The purpose of this work is to pursue an effective modification for PGA processing and application via melt blending.

## 1. Introduction

The problems of non-biodegradable “white pollution” and microplastics in the ocean have become more and more urgent and greatly promote the importance of the development and application of biopolymers [1,2,3]. As one of the biodegradable polymers, polyglycolic acid (PGA) demonstrates excellent mechanical performance, high crystallinity and fully-solved ability (or degradable) in seawater environments, thus it has currently become the most attractive biopolymer for the construction of a low-carbon society [4] and thus is widely used in many fields, such as surgical implants [5,6], packaging film [7] and oil/gas exploitation tools, etc., [8]. Under non-enzymic hydrolysis, water dissolution and composting conditions, the PGA can be decomposed by carbon dioxide and water which causes no pollution. With the rapid development of PGA polymerization technology, industrialized mass production is imminent. The Sinopec Group China has just finished its initial investment in factory construction for the mass production of PGA. Therefore, the applications of PGA will be definitely extended to consumer products with an accepted price in the future.

The molecular structure of PGA is similar to that of polylactic acid (PLA). Owing to the lack of side methyl, PGA material presents a higher crystallinity (45–55%) and an improved thermal stability, gas barrier property and remarkable mechanical strength (~115 MPa) as well as stiffness (~7 GPa). Therefore, PGA is a perfect candidate to alleviate the excessive consumption of fossil-derived plastics. It should be noted that the attractive mechanical performance of PGA and it water-soluble property endow PGA with the potential to replace the magnesium–aluminum alloy in the oil/gas exploitation field [9,10,11].

Although PGA has attracted wide interest as an environmentally friendly polymer, the inherent brittleness of PGA products and thermal instability of its melt above the melting temperature (*T*_m_) severely limit its applications to a great extent [12]. It was reported that the elongation at the break and the notch impact strength of pristine PGA were only 4.8% and 3.5 kJ/m^2^, respectively, whereas the tensile strength is as high as *ca.*117 MPa, which is much higher than that of nylon (PA), near to PEEK [12,13]. The *T*_m_ of PGA is ca. 220–230 °C, the crystallization temperature (*T*_c_) is ca. 180–200 °C and the glass transition temperature (*T*_g_) is ca. 35–40 °C. The disappointing thermal instability requires that the dwell time of the PGA melt in the barrel should be short. Meanwhile, the high *T*_c_ suggests that the temperature drop during the PGA melt flow should be low in case the melt cools and crystallizes.

In addition, the prominent water-soluble ability of PGA also promotes its applications in the field of petrochemical resource exploitation. The water-soluble ability of PGA is mainly attributed to hydrolysis and degradation in a water environment, which can be manipulated by blending PGA with a component which is easy to hydrolyze or dissolve in water [14,15,16,17]. It is well recognized that the poly(vinyl alcohol) (PVA) is a typical water-soluble polymer with a tailorable degree of hydrolysis with varied water-soluble abilities. However, the poor plasticizing property of PVA greatly limits its thermoplastic processing which normally need plasticizing modification. In addition, the unsatisfied thermal stability of PVA is also acknowledged owing to the dehydroxylation behavior at a high temperature. The melting temperature *T*_m_ of PVA is about 220–240 °C and its thermal degradation temperature *T*_d_ is 200–250 °C [18,19]. Therefore, PVA demonstrates a *T*_m_ close to that of PGA which provides the potential to melt blend them together, but it is more sensitive to thermal degradation [20,21,22,23]. According to the results proposed by Thomas et al. [23], the thermal degradation process of PVA would be affected greatly by the atmosphere and its crystallinity. The mechanism for its degradation in an inert atmosphere was in accordance with the mechanism of elimination followed by pyrolization. Provided by an air atmosphere, the degradation of PVA becomes oxidative. The first degradation step in air is the reduction in the O–H species, but it is not solely due to the elimination of water because the oxidation of the polymer is simultaneously occurring [23]. As the PVA polymer with the best water solubility, therefore, it is interesting to blend PGA and PVA together to explore its application [24]. It can be compounded with other biodegradable polymers to obtain water-soluble degradation characteristics and thus reduce microplastics particles in the ocean.

The brittleness of PGA can be economically improved by blending with elastomeric polymers or acrylic impact modifiers [12]. Thermoplastic elastomers, such as PBAT [25,26], poly (butylene succinate) (PBS) [27] and polycaprolactone (PCL) [28], are generally adapted to blend with PGA, thus they were commonly preferred for toughening the PGA. However, the incompatibility between PGA and the modifier always leads to phase separation and thus deteriorated the final mechanical performance. Therefore, interface modification could be the main work to deal with the incompatibility [29]. Among them, PBAT possesses an excellent ductility and was considered a good candidate to improve the flexibility of PGA, but the high water-degradation temperature limits its application as one of the water-dissolution components.

In order to take both the advantage of superior mechanical strength and the water-soluble property of PGA, high-water-solubility PVA and low-melting-point POE were employed to toughen and regulate the mechanical property and water dissolution of PGA. The typical compatibilizer EMA-GMA AX8900 was used to improve the interface adhesion among PGA/PVA/POE during the blending process. Then, the anti-compression property, rheological behavior, morphology and thermal stability of the blends were investigated. Particularly, the thermal stability of the as-prepared blends was studied via both isothermal shear and thermogravimetric analysis (TGA). The shear viscosity and modulus were measured under a constant temperature for dozens of minutes. Anti-compression measurements were performed to evaluate the hydrostatic pressure resistance of PGA blends, which was regarded as an evaluation of the mechanical performance of the blends. Finally, the water-soluble ability of the blended PGA composites was accessed by a water dissolution process at 80 °C for 12 h.

## 2. Experimental Details

### 2.1. Materials

PGA granules (*M*_w_ = 1.5 − 2.0 × 10^5^ g/mol, MFI = *ca.*50 g/10 min under 230–235 °C) were provided by Shanghai Pujing Chemical Industry Co., Ltd., Shanghai, China. Polyvinyl alcohol (PVA, EXCEVAL AQ-4104, KURARAY, Tokyo, Japan), polyolefin elastomer (POE, ENGAGE^TM^ 8180, Dow Chemistry, Midland, MI, USA) and epoxy functionalized copolymer (ethylene-methyl acrylate-glycidyl methacrylate) (EMA-GMA, LOTADER AX8900, Arkema Innovative Chemistry Co., Ltd., Shanghai, China) were purchased from domestic companies, respectively. The EMA-GMA was used as a reactive compatibilizer to improve the compatibility between PGA and PVA. The POE presents a melting point *T*_m_ of ~47 °C and a melt flow index of ~0.5 g/10 min. The PVA has a degree of hydrolysis ~97.5–99.0 mol%, which is a thermoplastic grade product and thus can be used for melt extrusion, blending and injection processing. Chain extender ADR 4468 (BASF, Ludwigshafen, German), antioxidant 1010 (MACKLIN, Shanghai, China) were adopted to assist the hot-melt blending of PGA, PVA and POE.

### 2.2. Sample Preparation

The raw PGA, PVA and POE materials were dried at 80 °C through a vacuum oven for at least 6 h before blending. Each kind of the blends (Table 1) was manually mixed after fully drying and vacuum packaged. The blends were prepared via one-step melt compounding using the batch mixer (Polylab-OS, Haake, Germany), under 235 °C and 100 rpm for 6 min. Chain extender ADR 4468, antioxidant 1010 and EMA-GMA were introduced into each group of blends via melt compounding. The function of EMA-GMA in interface modification was well reported in references [11,12,30,31]. All the components were mixed manually at first and then were put into the batch mixer for melt compounding. Then samples with a dimension size of 5 mm × 5 mm × 3 ± 0.5 (thickness) mm were fabricated for compressive testing.

### 2.3. Characterizations

The compression strength was measured using a universal testing machine (ESM 303, Mark-10, New York, NY, USA) with a 1500 N load sensor (Mark-10, New York, NY, USA) at a velocity of 13 mm·min^−1^. Each test was repeated at least three times and the curve with the median value was used to represent the stress–strain behavior. The cross-sectional morphology was examined via the scanning electron microscopy (SEM, Hitachi SU8010, Tokyo, Japan) at an accelerating voltage of 3–5 kV. Differential scanning calorimetry (DSC-Q2000, TA Instruments, New Castle, DE, USA) was employed to examine the thermodynamic behavior of blends. The temperature of the specimen was increased from 30 to 250 °C with a heating rate of 10 °C/min, followed by cooling to 30 °C at a rate of −10 °C/min. TGA behaviors of PGA and PGAX in nitrogen atmosphere were measured using a thermogravimetric analyzer (TG-DSC-EGA, SETARAM Instrumentation, Lyon, France) within 25–500 °C at a rate of ±10 °C/min. Rheological behavior of the blends were measured by using a rotational rheometer (MCR 302, Anton Paar, Graz, Austria) under varying temperatures and shear frequency of 0.01–100 rad/s. In addition, isothermal constant shear behavior of PGAX and PGA melts were also measured with the duration of 30 min, which was regarded as a measure of thermal stability of samples in this work. The evaluation of water dissolution ability was performed using a beaker which was placed in an oil bath at 80 °C. Granular blends were immersed in 400 mL water with the help of magnetic stirring. After being dissolved for 12 h, the sample was taken out of water and was further dried completely. The residual mass fraction, the residual weight divided by the original weight, was proposed as the index to indicate the water dissolution ability for each sample.

## 3. Results and Discussion

### 3.1. Morphological Characterization

Morphological structures of the as-prepared PGAX blends were detected using SEM, as shown in Figure 1. An incompatible characteristic between PGA and PVA is clearly shown from the SEM observations. The minor-mass-fraction PVA was randomly dispersed within the PGA matrix at a microscopic scale (Figure 1a–f). The dimension size of the exposed PVA particles which are demonstrated as individual spheres is less than 5 μm, indicating a good dispersion state owing to the shear effect. In addition, the size of PVA spheres increases with the increase in the PVA content, raised from less than 1.0 μm in PGAX1 (Figure 1a) to 2–3 μm in PGAX5 (Figure 1e). The interface boundary between PVA and PGA is clearly observed, implying a phase separation. Meanwhile, the existence of POE seems to facilitate the dispersion of PVA, as reflected in Figure 1f. The cross-section morphology of PGAX6 reveals much smaller PVA spheres and a smoother fractured surface than those demonstrated in PGAX5.

In order to explore the compatibility degree, the thermal kinetics of PGAX blends was investigated by DSC, as shown in Figure 2. The presence of PVA not only decreases the melting temperature of PGAX, but also expands the melting temperature range. For instance, the melting temperature range of pristine PGA is ~213–223 °C while it becomes 195.5–212.7 °C for PGAX6. More significant endothermic peaks for the samples of PGA and PGAX1 are observed at the temperature of 220–223 °C, indicating a higher degree of crystallinity. This means that the addition of PVA damages the crystallization kinetics. In addition, the increased temperature difference between *T*_m_ and *T*_c_ reveals the expansion of the processing temperature range.

Furthermore, the presence of POE seems to hinder the crystallization kinetics, which is suggested by the lower crystallization temperature in Figure 2b. For example, the crystallization temperature of PGAX4 is obviously lower than that of PGAX3. When the PVA content is raised up to 30 wt%, the crystallization enthalpy at 175–181 °C decreases greatly. In addition, two crystallization temperature peaks are observed for PGAX6, indicating double crystallization morphology. The endothermic process reveals similar patterns, implying the absence of chemical reaction during melt compounding. However, the melting temperature range is substantially expanded with the increased PVA content, e.g., PGAX6, and the processing temperature could be set at 200–210 °C.

### 3.2. Anti-Compression Properties

The anti-compression strength of the blends was measured as an index to access their mechanical performance. The testing method adopted here is not typical or standard, but is beneficial to understanding the development and the applications of PGAX blends. The compressive measurements were performed by a self-designed gauge. Figure 3 shows the compressive load–displacement patterns of all the PGAX blends during anti-compression testing. It can be found that the highest anti-compression strength was revealed from the pristine PGA sample. A linear response of load versus travel (i.e., the compression displacement) is revealed and the PGA sample maintained well, indicating a high compression modulus and stiffness. This result also confirms the excellent mechanical properties of PGA material. With the presence of PVA, however, the ultimate compressive load decreases greatly and the sample is found to be easily crushed (Figure 3b,c), behaving a much more remarkable fragile characteristic than that of PGA (Figure 3b,c), which suggests a decreased compression modulus. Specifically, the pristine PGA can withstand a compressive force of up to 1100 N without being damaged, whereas the PGAX4 only shows an ultimate load of ~660 N. The decreased compressive strength of the PGAX blends should be attributed to the incompatibility between PVA and PGA, which causes a strong stress concentration effect around the interfacial region and usually requires interfacial modification [12,32]. Meanwhile, the presence of POE also decreases the compressive strength owing to the low modulus itself. For instance, the ultimate compressive loads of PGAX2, PGAX4 and PGAX6 are lower than those of PGAX1, PGAX3 and PGAX5, respectively.

With the content of PVA being increased, however, PGAX5 and PGAX6 demonstrate a reverse change in their compression ability. When 46.2 wt% PVA compounded with 30.7 wt% PGA, the continuous phase within PGAX 6 changes to PVA instead of PGA. Then the PGAX6 behaves as a mechanical property which is similar to PVA and affected simultaneously by POE. In this case, the brittle fracture is hardly found from the PGAX6, which means an enhanced compression capacity. Consequently, the similar compressive behavior of PGAX6 with that of pristine PGA was observed. Firstly, there is no doubt that the presence of PVA definitely decreases the mechanical strength of PGA owing to the incompatibility between them. Second, the use of POE must further decrease the mechanical strength. Hence, we can infer that the obtained compressive property of PGA6 should attribute to the induction of a large number of interface defects, endowing the PGAX6 with a porous structure. Such a transition from brittleness to toughness ensures the PGAX6 can endure a similar compressive load as PGA.

### 3.3. Rheological Behaviors

In order to obtain an insight into the processing ability of PGA and the PGAX blends, the rheological behaviors of PGA and PGAX melts under varying temperatures were measured with the oscillation shear mode, as shown in Figure 4 and Figure 5. The complex viscosity *η*, loss modulus G″ and storage modulus *G*′ of the PGA melt at 220–240 °C within a shear frequency of 0.01–100 s^−1^ were presented. Overall, the viscosity of the PGA melt demonstrated a clear dependence on temperature, i.e., decreasing with the increase of melt temperature. At the shear frequency of 100 s^−1^, almost all the measured shear viscosities of the pristine PGA melt were below 20 Pa·s, and it decreased from 11.6 Pa·s at 220 °C to 2.6 Pa·s at 240 °C. However, the viscosity value kept relatively stable until the shear frequency decreased to 1.0 s^−1^, revealing a typical Newtonian plateau. This means that the PGA melt has a very good flow ability when its temperature rises up to *T*_m_, suggesting that increasing the shear rate has little effect on reducing its viscosity. The shear-thinning behavior disappears when the shear rate is above 0.1 s^−1^. The complex shear viscosity at 240 °C and 100 s^−1^ is about 2.6 Pa·s, while it increases to 181.4 Pa·s at 0.01 s^−1^. For a pure PGA melt, therefore, a constant viscosity would be achieved under a certain temperature. The low shear viscosity of PGA endows a low melt strength, thus it is usually required to be enhanced [33].

The *G*′ and *G*″ behave with quite different behaviors with the shear rate, as shown in Figure 4b,c. The *G*″ of PGA at certain temperature increases with the shear rate, while it decreases with the increase of the melt temperature. As to *G*′, however, the results show that the measurements could be instable. The fluctuation of the *G*″ value is very obvious and remarkable. This reveals that the elasticity of the pristine PGA melt is rather low or even undetectable.

The rheological behaviors of the PGAX blends are shown in Figure 5. The complex viscosity of PGAX increases with the increase in the PVA content and the shear-thinning behavior is much more remarkable. This implies that both the presence of PVA and POE greatly enhance the elasticity of the PGA melt. The measured viscosity increases from ~4.4 Pa·s for PGAX1 to 227.5 Pa·s for PGAX6 under 100 s^−1^. The *G*′ is found to be increased by 3–4 orders of magnitude and the *G*″ is increased by 2–3 orders of magnitude. The *G*″ decreases slightly with the increase in the shear frequency until ~0.1 s^−1^ and then increases. In the practical measuring process, the initial shear frequency was 100 s^−1^ and then decreased gradually to 0.01 s^−1^. The whole test process took about 45 min. Therefore, the slight increase of *G*″ from 0.1 to 0.01 s^−1^ could be resulted from the thermal degradation of PGAX. Figure 5d further confirms the thermal degradation of PGAX. The Cole–Cole plots of the PGAX1-5 melts exhibit a reversal pattern, whereas the PGAX6 demonstrates a Cole–Cole pattern coming close to the isomodulus line (*G*″ = *G*′) which suggests a strong elastic response to the oscillation shear. Such a reversal pattern implies that the melt samples could be unstable during the testing process.

### 3.4. Thermal Stability

The thermal stability of a polymer usually determines the suitable strategy of the processing optimization. It is well acknowledged that the PGA presents poor thermal stability, for which thermal degradation usually occurs at a temperature just above its *T*_m_. The temperature of the peak thermal-degradation rate of PGA is around 250–255 °C, which is near its melting point of ~220 °C [34]. In this work, the thermal stability of PGA was investigated through isothermal shear rheology and TGA analysis, as shown in Figure 6. The oscillated shear viscosity (Figure 6a), loss factor (Figure 6b), storage/loss modulus (Figure 6c) at 225 °C and 10 rad s^−1^ and TGA behavior (Figure 6d) were all considered to evaluate the thermal stability of PGA. Within 150 min, the complex shear viscosity slightly reduced from 6.5 to 5.5 Pa·s and the value of tan*δ* decreased from ~20 to ~2.4. The modulus variation in terms of time indicates that the molecular entanglement degree could be varied with shear time. The *G*″ and *G*′ curves demonstrate linear decrease and increase patterns within the shear period, respectively. This implies an occurrence of thermal degradation during the testing process, although this cannot be detected by TGA analysis (Figure 6d). Figure 6d shows the thermal stability of pure PGA under a nitrogen atmosphere. It clearly suggests that the onset of the thermal degradation of PGA is above 250 °C, which was consistent with the references [34,35]. This implies that the thermogravimetric behavior can only be observed above 250 °C, but this does not mean the thermal stability of the PGA melt is below 250 °C. The thermal degradation of PGA includes the breakage of the ester bond, leading to a decrease in the molecular weight and an increase in the molecular weight distribution. However, weight loss is hardly observed during such a process owing to the lack of VOCs. It is confirmed that the thermal degradation of PGA occurs exactly just above its *T*_m_, i.e., ~220 °C.

Similarly, the isothermal shear behavior of the PGAX melts at the conditions of 225 °C and 10 rad s^−1^ was measured within a 30 min period, as shown in Figure 7. The PGA melt presented a relatively stable shear behavior, for which the shear viscosity was kept almost constant for 30 min. However, the shear viscosity of PGAX displayed an increasing trend within the time range. Moreover, the viscosities of the PGAX blends were significantly increased by the addition of PVA. A higher shear viscosity was observed for the PGAX with a higher PVA content. However, the difference between these viscosity values tends to be reduced with the time, which means that the phase morphology of the PGAX blends could be varied with time. On the one hand, the chain extender ADR 4468 is contributive to enhancing the melt strength, thus increasing the viscosity of the PGA melt [33]. On the other hand, thermal degradation of the PVA occurs, which further decreases the flow ability of its melt. The onset of the thermal degradation of PVA is about 220 °C, which means its thermal degradation happens throughout the whole testing process [23,36,37].

Furthermore, thermal gravimetric analysis was also conducted for PGAX blends, as shown in Figure 8. It is interesting that the thermal weightlessness behavior occurs at the beginning of 50 °C, near 1% weight loss being observed at 50 °C for all the PGAX samples and, particularly, ~8% weight loss for PGAX6. Meanwhile, the degree of the thermal weightlessness of PGAX2, PGAX4 and PGAX6 are higher than those of PGAX1, PGAX3 and PGAX5 within the temperature of 50–200 °C, respectively. This means that the use of PEO is unfavorable for the enhancement of the thermal stability of the PGAX blends, whereas it promotes the thermal degradation of PGAX under a low temperature.

Unlike the thermal stability behavior of PGA, the thermal instability of PGAX, which was observed at a relatively low temperature, could be attributed to the melt blending process, during which the thermal degradation of PVA occurred. Results displayed in Figure 8 also remind us that the thermoplastic processing conditions of PGAX should be rather critical. Thermal degradation in such a processing is almost inevitable, so reducing the residence time of the PGAX melt should be considered as a solution.

### 3.5. Water-Soluble Ability

Owing to the presence of PVA, the blends of PGAX exhibited distinct water-soluble behaviors with the varying content of PVA and POE. Figure 9 compares the water-soluble efficiency of PGAX under 80 °C within 12 h. The mass ration was calculated by dividing the residual weight by the original weight of a sample. The pristine PGA presents a water-dissolution ratio about 15% under 80 °C (provided by the manufacturer), so 12 h is sufficient to indicate the difference of the water-soluble ability of the blends.

Overall, the presence of PVA and POE is beneficial to promoting the water-soluble ability of PGAX blends, but the effect of PEO on the water-soluble ratio is varied when the PVA content is up to 46.2 wt%. The lowest residual mass ratio is observed from PGAX5. This is because the weight content of PVA is higher than that of PGA. Furthermore, the residual mass ratio is relatively raised to near 80% for PGAX6. This means the PEO protects the PVA and PGA from water dissolution, which can be implied from the morphology variation, as shown in Figure 10.

The morphological difference can be clearly observed by comparing Figure 1 and Figure 10. For a single PGA particle, the water-soluble process is a typical dissolution process from outside to inside. After water dissolution, the surface of PGA shows many uniform but irregular grooved caves which are mainly at the micron scale. The whole sample could be broken up to small pieces. For the PGAX blends, however, the PVA will dissolve at first as the sacrificial phase and consequently promote the water-soluble process of PGAX. The water-soluble efficiency mainly depends on the contact area between the water and water-soluble components, i.e., PVA and PGA. The PEO was melted at 80 °C and then transferred into viscous flow state, but it was not really dissolved in water. Nevertheless, the dissolved PVA and PEO provide a greater interface area between PGA and water and thus promote the water dissolution process. Interestingly, the PGAX6 illustrates a reversed characteristic, of which the water-soluble ratio is lower than that of PGAX5. The PEO could hinder the water dissolution process of PVA and PGA by blocking their contact with water, but the dissolution ratio of PGAX6 is still largely enhanced.

The morphology of PGAX6 after water dissolution was found to be different from those of PGAX1-5. Neither individually dissolved components, such as PVA, nor an irregular gully-shaped structure can be observed within PGAX6, whereas the state of its dissolution demonstrates a uniform process. Owing to the dissolution of PVA and POE, the viscous-paste state of PGAX6 was observed after being waster-dissolved, indicating the complete loss of mechanical strength.

## 4. Conclusions

In summary, the mechanical, thermal, rheological and water-soluble properties of PGA/PVA blends were investigated with varying PVA contents, ranging from 15.4 to 54.5 wt%. The PGA/PVA blends demonstrated improved viscoelasticity and water-soluble efficiency while the anti-compression strength and thermal stability were deteriorated. The existence of POE induced a decrease in the compression modulus of the PGA/PVA blends, leading to a reduced ultimate compressive force at the fragile point. The compression limitations for the samples with the presence of PVA were observed ranging ~1000–~700 N, whereas the pristine PGA demonstrated a linear compressive behavior without the occurrence of being crushed. In addition, the complex viscosities of PGAX blends were largely increased and their thermal stability was deteriorated correspondingly by introducing PEO into the blends. The pristine PGA melt revealed a shear viscosity of only ~10 Pa·s at the shear rate of~1.0 s^−1^, while the viscosity of PGAX was raised up to ~75, ~191, ~412, ~930, ~2867 and ~2612 Pa·s for PGAX1-6, respectively. The obtained isothermal shear behaviors of these PGAX blends clearly indicated that the PVA would largely decrease the thermal stability of PGA, causing the onset of PGAX blends at a relatively low temperature. Owing to the incompatibility between PVA and PGA, the PVA was found to be randomly dispersed within the PGA matrix and played the role of the sacrificial water-dissolution component. The lowest water-dissolution ration was revealed from the PGAX5, showing a residual mass ratio of only ~58%. For the PGAX6, however, the PEO shows its advantage on hindering the water dissolution ability, leading to an increased residual mass ratio; however, the high concentration of PVA could result in the loss of mechanical properties. This work has demonstrated a potential strategy to regulate the mechanical and water-soluble properties of PGA by blending with PVA and PEO, and provides a routine to study the thermal degradation of those temperature-sensitive materials.

## Figures and Tables

**Figure 1 polymers-14-03375-f001:**
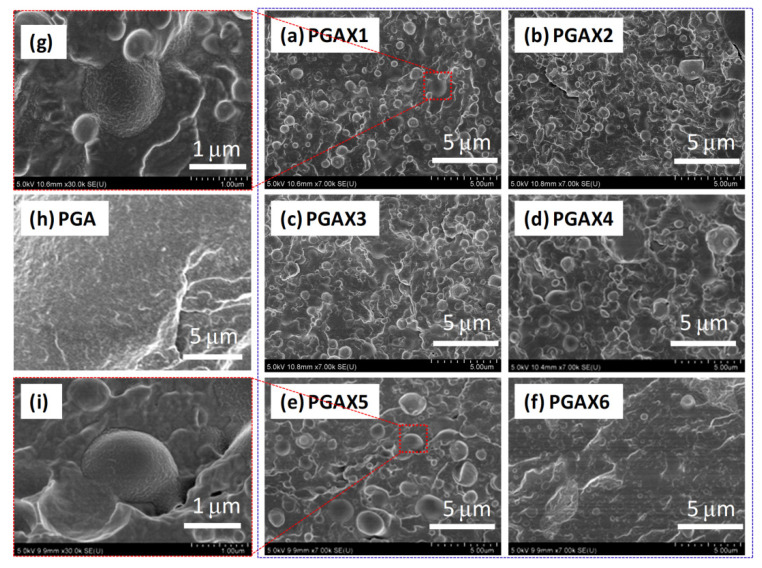
SEM images of the compounded PGAX blends (**a**–**f**) with magnification in (**g**,**i**) and the pristine PGA (**h**).

**Figure 2 polymers-14-03375-f002:**
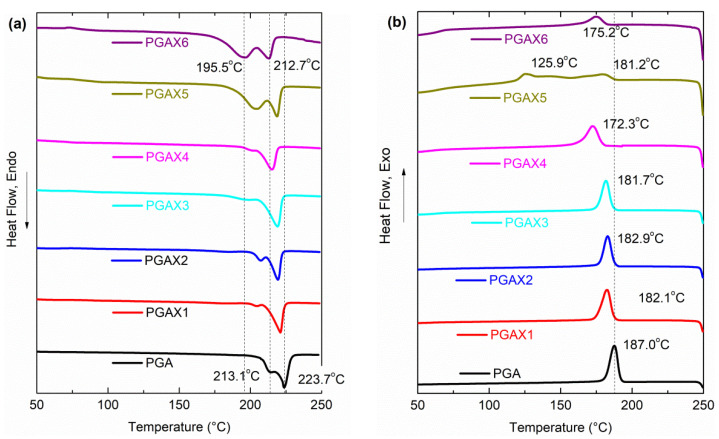
DSC patterns of PGA and PGAX: (**a**) DSC heating curves and (**b**) DSC cooling curves.

**Figure 3 polymers-14-03375-f003:**
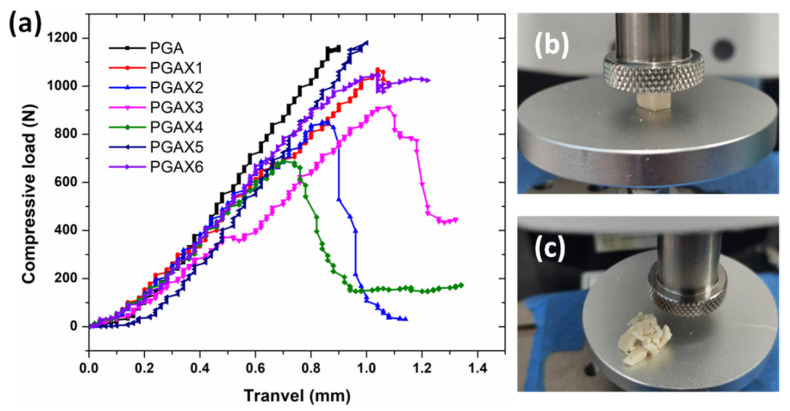
Anti-compression properties of PGA and PGAX blends: (**a**) compressive curves, (**b**) the PGAX3 sample before and (**c**) after compression.

**Figure 4 polymers-14-03375-f004:**
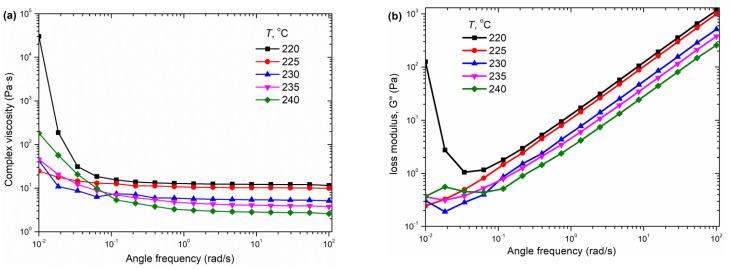
Rheological properties and DSC patterns of pristine PGA melt: (**a**) complex viscosity, (**b**) loss modulus, (**c**) storage modulus and (**d**) the DSC curves of PGA.

**Figure 5 polymers-14-03375-f005:**
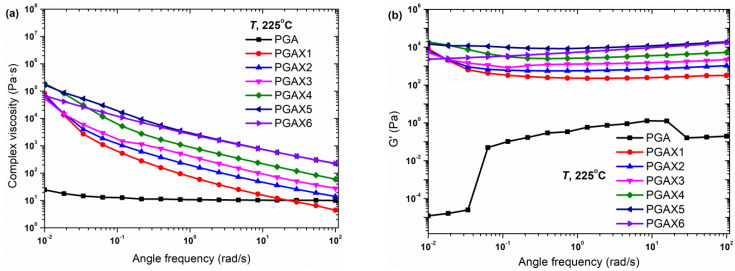
Rheological properties of compounded PGAX composites: (**a**) complex viscosity, (**b**) storage modulus, (**c**) loss modulus and (**d**) Cole–Cole plots.

**Figure 6 polymers-14-03375-f006:**
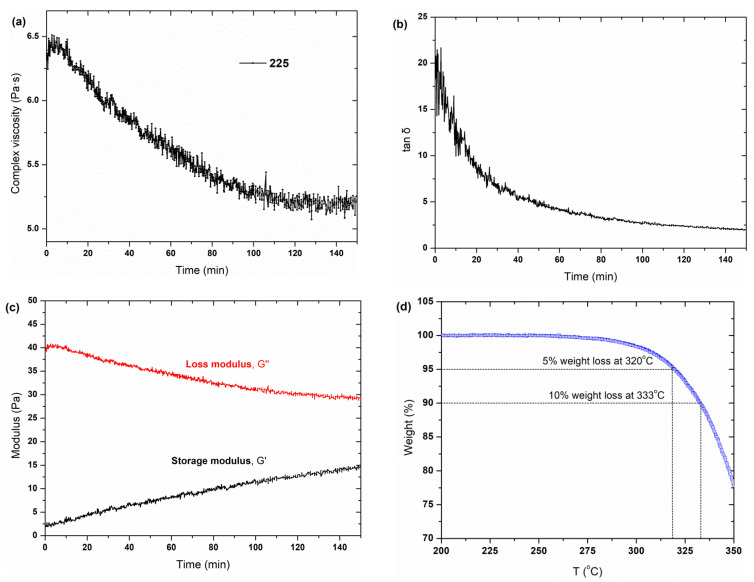
Measurements of the thermal stability of pristine PGA: (**a**–**c**) isothermal shear behavior under 10 rad s^−1^ and (**d**) TGA analysis within 200–350 °C.

**Figure 7 polymers-14-03375-f007:**
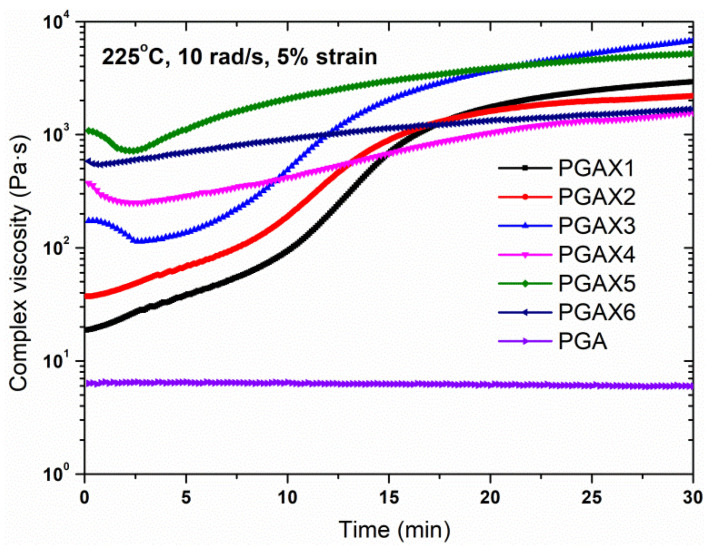
Isothermal shear viscosity of PGA and PGAX composites under 225 °C within 30 min.

**Figure 8 polymers-14-03375-f008:**
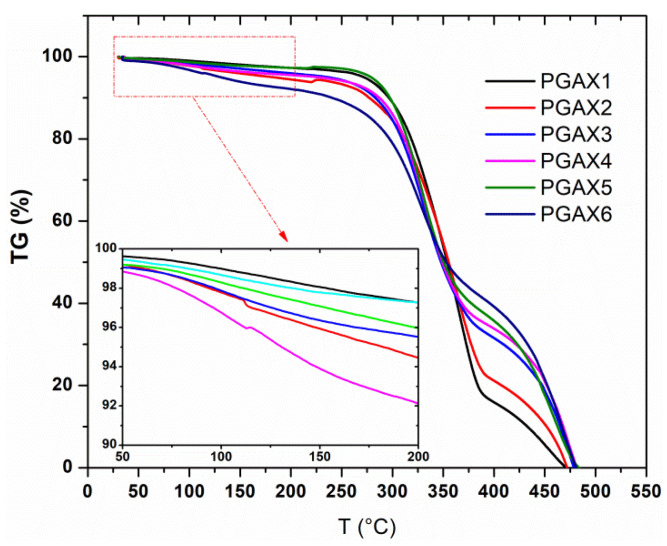
TGA analyses of PGAX blends under N_2_ atmosphere.

**Figure 9 polymers-14-03375-f009:**
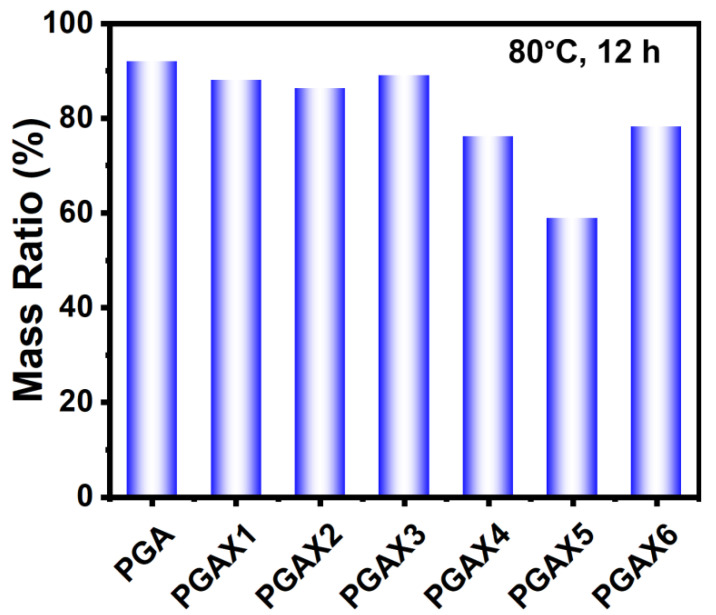
Residual mass ratio of PGA and PGAX blends after water dissolution at 80 °C for 12 h.

**Figure 10 polymers-14-03375-f010:**
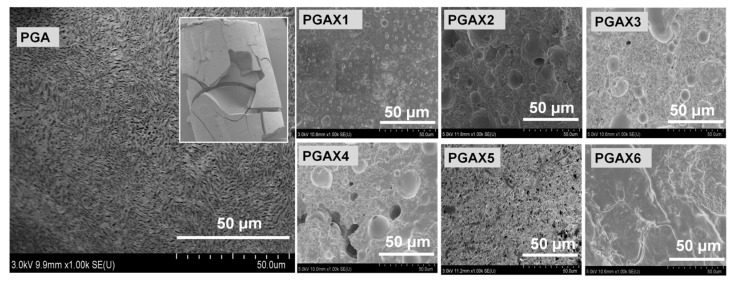
Surface morphology of PGA and PGAX1–6 after water dissolution process in this work.

**Table 1 polymers-14-03375-t001:** Formulations of samples for batch compounding.

Samples	Components			
	PGA (wt%)	PVA (wt%)	POE (wt%)	EMA-GMA (wt%)
PGAX1	72.7	18.2	0	9.1
PGAX2	61.5	15.4	15.4	7.7
PGAX3	54.5	36.4	0	9.1
PGAX4	46.2	30.7	15.4	7.7
PGAX5	36.4	54.5	0	9.1
PGAX6	30.7	46.2	15.4	7.7

## Data Availability

Not applicable.

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
