# Peer review of "Modifying Anti-Compression Property and Water-Soluble Ability of Polyglycolic Acid via Melt Blending with Polyvinyl Alcohol"

_polymers, 2022, doi:10.3390/polym14163375_

Round 1
Reviewer 1 Report
The manuscript “Modifying anti-compression property and water-soluble ability of polyglycolic acid via melt blending with polyvinyl alcohol” presents a study of binary blend PGA and PVA using a compatibilizer and a thoughening agent. The purpose of the study is to analyze the thermal stability of PGA, anti-compression properties as well as the application of PGA via melt blending. Their results showed the imcompatibility of PGA-PVA blends, a low compressive load of blends. The study is interesting and well discussed, I just have a couple of questions
1. Is important to indicate at the abstract what is the effect to use a Compatibilizer and the thoughening agent, also along the manuscript.
2. The formulation 2, 4 and 6 use a 78 gr and 1, 3 and 5 use 66 gr. When you prepare the blend in the internal mixer, Do you think there is a mixing effect?
3. What type of sample was used in the water solubility tests? Do you consider that a certain portion of the soluble part could be encapsulated with the non-soluble part?
4. What type of application is suggested for these formulations?
Reviewer 2 Report
In title, first character of each word should be capital. Please follow journal guidelines.
Who is corresponding author indicate, it as per journal guidelines.
In abstract background sentence is missing.
In abstract:
Please write polyvinyl alcohol as poly (vinyl alcohol) in the manuscript.
“In this paper, binary blends including varying contents of polyglycolic acid (PGA) and polyvinyl alcohol (PVA) were prepared via melt compounding with the help of the (ethylene-methyl acrylate-glycidyl methacrylate) (EMA-GMA) as compatibilizer and the polyolefin elastomer (POE) as toughening agent”. Please make it short and write preciously.
“ .. …. PGAX samples with 10-20 wt.% PVA” in the manuscript nowhere mention 10-20wt%. Remove dot mark before %.
In introduction:
‘, just closing to PEEK [4, 5]’, re write this sentence by replacing just closing.
Introduction need to rewritten more deeply with recent literatures (2021, and 2022).
Describe advantages melt blending of PVA.
Very less references in the manuscript increase it at least 40.
“The water-soluble ability of PGA is mainly attributed to hydrolysis and degradation in water environment, which can be manipulated by blending PGA with a component which is easy to hydrolyze or dissolve in water [6]. ‘It is well recognized that the polyvinyl alcohol (PVA) is a typical water soluble polymer with tailorable alcoholysis degree with varied waster-soluble abilities’ This sentence need citations, insert it Carbohydrate Polymers, 2021, 257, 117633, https://doi.org/10.1016/j.progpolymsci.2009.05.003 and https://doi.org/10.1007/s10924-022-02454-w. What is “alcoholysis”. check right terminology.
However, the poor plasticizing property of PVA greatly limits the hot-melt processing and thus need to be plasticized. In addition, the unsatisfied thermal stability of PVA is also acknowledged owing to the dehydroxylation behavior at high temperature. The melting temperature Tm of PVA is about 220-240°C and its thermal degradation temperature Td is 200-250°C. This sentence need citations, insert it Polymer Engineering and Science 62(5) (2022) 1526 and Food packaging and Shelf Life 33 (2022) 100904.
In other place the following references are suitable cite it accordingly Desalination, Journal of Membrane Science,
Describe why melt blending of PVA advantages.
In section PVA molecular weight degree of hydrolysis information is missing.
In sample preparation all the quantity of raw materials can authors covert to wt%.
There is no characterization or measurement part. Please include in section wise.
Please add the interaction scheme between raw materials.
How authors confirm interaction there is no such data in the manuscript. Please include it.
Fig 2 what is a and b mention clearly in caption. Also check in fig 5,4 and 10.
Insert references papers in each result and discussion to support your conclusions.
What is mechanism of anti-compression activity. Please mention it clearly. Section 3.2 need to be rewritten.
Fig. 6 d is not clear.
In TGA analysis please provide the quantative data.
Calculate the degree of crystallinity from DSC curves. Fig. 2.
How authors calculate the mass ratio there is no formula. Mention it. Why authors tested 80oC and 12 hr. 12 hr is so short time. At least need to do 72 hrs. Which standard method authors follow. Please mention it.
How about mechanical properties of composites.
In conclusion section please mention quantative data and rewrite it accordingly preciously.
Check grammar throughout the manuscript.
Round 2
Reviewer 2 Report
The authors improved the manuscript.